# Differential Susceptibility or Diathesis-Stress: Testing the Moderating Role of Temperament and Cortisol Levels between Fathers’ Parenting and Children’s Aggressive Behavior

**DOI:** 10.3390/brainsci11081088

**Published:** 2021-08-19

**Authors:** Eider Pascual-Sagastizabal, Nora del Puerto-Golzarri, Aitziber Azurmendi

**Affiliations:** Department of Basic Psychological Processes and Their Development, Faculty of Psychology, University of the Basque Country (UPV/EHU), 20018 San Sebastian, Spain; eider.pascual@ehu.eus (E.P.-S.); aitziber.azurmendi@ehu.eus (A.A.)

**Keywords:** aggressive behavior, cortisol, temperament, parenting styles, children

## Abstract

Aggression is a multidimensional behavior that could be caused by different biopsychosocial variables. The aim of this study was to explore whether temperament, cortisol and sex moderate the relation between fathers’ parenting style and aggressive behavior in school-aged children, and whether this corresponds to differential susceptibility or diathesis-stress. Participants were 158 children (88 boys and 70 girls) aged 8 years. Aggressive behavior was measured using the Direct and Indirect Aggression Scale and fathers informed about their child’s temperament and their own parenting style through the Children’s Behavior Questionnaire and the Parenting Styles and Dimensions Questionnaire (respectively). Children’s’ baseline saliva cortisol levels were analyzed through an enzyme immunoassay technique. The results revealed that high cortisol levels moderated the relation between fathers’ low levels of authoritative parenting and boys’ aggression. Moreover, high negative emotionality moderated the relation between permissive paternal parenting and girls’ aggressive behavior, with both these relations being consistent with the diathesis-stress theory.

## 1. Introduction

Aggressive behavior is a multidimensional phenomenon involving numerous physical, emotional, cognitive and social factors [1]. Children’s aggressive behavior is a social problem, which is causing increasing concern both in society in general and in relation to public health [2]. Moreover, extreme forms of aggression are one of the main causes of death and the WHO [3] declared that by 2030 it may take more lives than very prevalent diseases. That is why prevention of aggression is very important at early stages of life. As Scott et al. [2] point out in their review, aggressive behavior should be approached from a biopsychosocial perspective, with special attention being paid to those factors that may predict it during early development phases, with the aim of avoiding negative consequences later on.

In relation to models, which aim to explain the different individual trajectories of behavior, they differ as regards their response, with some people being more sensitive to certain factors under the same environmental conditions than others [4]. Those models have shown an increasing importance over recent years as all human beings are affected by different aspects of their social and physical environment.

The diathesis-stress model, which associates personal vulnerability with risk contexts, has been considered particularly useful by some authors [5,6] for explaining aggression. According to this model, aggressive behavior emerges as the result of the interaction between a biological predisposition (genetic, physiological, temperamental, etc.) and stressful life events. Nevertheless, based on theories of biological sensitivity to context and differential susceptibility, other authors [7,8] suggest that some individuals are more susceptible than others to both negative (risk-promoting) and positive (development-optimizing) environmental conditions leads to for better and for worse outcome related to aggressive behavior. These individuals are considered to be “plastic” or malleable, since they adapt to their environment [9].

Of the different characteristics that have been proposed as susceptibility or vulnerability factors, the present study will focus on children’s temperament and cortisol levels taking into account sex differences. These variables have been shown by prior research to be involved in aggressive behavior.

Temperament can be defined as innate individual differences in people’s behavioral and emotional tendencies [10]. Several studies have established an association between a range of temperament characteristics and aggressive behavior in both childhood and adolescence [11,12]. Of the different components of temperament, negative emotionality, consisting of anger, fear and frustration, among others, is believed to predict externalizing and internalizing problems [13,14,15]. For its part, another characteristic, effortful control, understood as the ability to regulate one’s attention, emotions and behavior [15,16], is considered to be a protective factor against the development of aggressive behavior [17,18]. Finally, surgency, which includes aspects related to an absence of shyness as well as to impulsiveness and high-intensity pleasure, seems to be associated with externalizing problems [14,19]. Many studies have found that children with one or some of these characteristics demonstrate more externalizing behavior problems or reactive aggressive behavior than their counterparts without these susceptibility characteristics if they have experienced poor parenting. They also demonstrate fewer behavioral problems when brought up in a positive environment, a finding which fits the differential susceptibility model [11,20,21,22,23]. Nevertheless, the results reported by other studies are more consistent with the diathesis-stress model [24,25]. The interaction between child temperament and family features assessed in late childhood and its association with adolescent internalizing and externalizing symptoms are also consistent with the diathesis-stress model [26]. Poorer parenting and family functioning were found to predict higher levels of internalizing and externalizing problems among youth with difficult temperamental features [27]. Individuals who were highly negative as infants reported more externalizing behaviors if they experienced low-quality childcare, but not fewer problems if they experienced high-quality care relative to their counterparts with less difficult temperaments in infancy [28].

In terms of hormones, cortisol has attracted a certain degree of attention within the field of susceptibility or vulnerability models. This glucocorticoid hormone is synthesized in the adrenal cortex as the result of the activation of the hypothalamic-pituitary-adrenal (HPA) axis and is linked to the regulation of diverse biological processes, helping the body respond to the demands of environmental conditions [29]. Many different studies have related cortisol to aggressive behavior, with some finding a negative relationship [30,31] and others a positive one between levels of this hormone and aggression in adulthood [32,33] and prepubescence [34,35]. One of the models proposed to explain this relationship posits that cortisol reactivity acts as a moderator between parental behaviors and aggression [36]. Some studies have found that children with high cortisol reactivity brought up in negative family environments have higher levels of externalizing and internalizing behaviors than their counterparts with low cortisol reactivity [37,38]. Others report that adolescents with high cortisol reactivity have more or fewer behavioral problems depending on whether they come from a stressful or non-stressful family environment [39]. Nevertheless, basal cortisol levels have hardly been taken into account as a susceptibility or vulnerability factor which makes its study in relation to aggression interesting [34,40].

The study of sex differences in aggressive behavior is another key element in this field. Diverse studies have demonstrated that, from a very early age, boys display much more aggressive behavior (particularly physical aggression) than girls [41,42,43]. The evidence suggests that this difference is not just the result of the gradual socialization process to which all humans are subject [41], but rather, may well be due also to biological factors [44]. In this sense, some studies have found significant interactions between biological and environmental factors in boys, but not in girls [45,46], although it is also true that other studies have found stronger environmental effects in girls [47], suggesting that the question still requires further study.

One key factor in all the models outlined above is environment or context. Psychobiological plasticity or vulnerability only has an influence in adverse and/or advantageous contexts. Of the diverse contexts analyzed, family context, and specifically parenting style, is one of the most widely-studied. Following Baurmind’s theory [48] there are three parenting styles. The authoritative parenting style is characterized by high levels of support and control, with parents being both warm and demanding. The authoritarian style, on the other hand, is characterized by low levels of support and a high degree of control, with parents demonstrating little warmth and exerting strict control. Finally, the permissive style is characterized by high levels of support and warmth, but low levels (or the complete absence of) control and demandingness). There is evidence to suggest that children reared in the authoritative style are emotionally more stable and have adaptive coping patterns [49,50]. The authoritarian and permissive parenting styles, on the other hand, have been linked to higher levels of aggressive behavior among children [51,52].

Most research has only focused on the study of mother’s parenting style as traditionally it is the mother who assumes the main responsibility for care and rearing tasks and fathers are considered helpers to the mother. Moreover, some studies only include data from the mother and assume that fathers behave similarly [53] although fathers themselves have an important role in the development of their children [54] and are becoming more involved in the care and rearing activities [55]. In this line, some authors have demonstrated the importance of studying the father’s parenting role effect on their children’s behavioral problems [23,56].

In light of the above, the present study aimed to analyze whether variation in family context (adverse-favorable father’s parenting) differentially predicts aggressive behavior in eight-year-old children in accordance with their temperament, cortisol levels and sex. We expected to find that: (a) low effortful control and high negative emotionality and surgency predict higher levels of aggressive behavior when coupled with an adverse family context (high permissive and authoritarian and low authoritative father’s parenting styles) and lower levels of aggressive behavior when coupled with a favorable family context (low permissive and authoritarian and high authoritative father’s parenting styles) in comparison with children who do not demonstrate these characteristics; (b) high levels of cortisol in an adverse family context predict more aggressive behavior, while in a favorable context they predict less aggression in comparison with children with low cortisol levels; (c) there will be sex differences in the context-temperament and context-cortisol interactions.

## 2. Materials and Methods

### 2.1. Participants

Participants were 158 eight-year-old children (88 boys and 70 girls) from four schools in San Sebastian, Northern Spain. The families of the 158 children were informed of the study and gave their written consent for their sons and daughters to participate in the study. The socioeconomic status of the sample was medium and medium-high.

All the tests were non-invasive, some of them were carried out in the children’s classrooms at 9 am by qualified members of the research team and other ones were parent reported. The project was pre-approved by the Ethics Committee of the University of the Basque Country both the Ethics Committee of Research Involving Human Beings (CEISH) (Protocol Code: CEISH 19_2010 and date of approval: 25 March 2010) and the Ethics Committee for Research Involving Biological Agents and GMOs (CEIAB) (Protocol Code: CEIAB 20_2010 and date of approval: 25 March 2010) and the procedure followed the pertinent national legislation.

### 2.2. Aggressive Behavior Measure

Aggressive behavior was measured using the Spanish version of the Direct and Indirect Aggression Scale (DIAS) [57]. This instrument is a peer estimation scale that assesses physical, verbal and indirect aggression through 24 items (for example: Does he/she hit a lot? Does he/she shut others out of the group?) on a five-point Likert-type scale (“never”, “seldom”, “sometimes”, “quite often” and “very often”). Due to the social segregation (sex-based segregation) typical of this age group, peer estimation was performed in accordance with sex, i.e., children were only asked to evaluate their same-sex classmates. This is because asking children questions about the opposite sex may give rise to estimation errors due to prejudice or lack of knowledge.

The final score for each child was obtained by taking the average of all the scores given by their same-sex classmates. Given the close correlation which exists between physical, verbal and indirect aggression (r ≥ 0.97), a global averaged aggression score was calculated for each child based on the scores obtained on all three. The reliability coefficient for this questionnaire was α = 0.97 in our sample.

### 2.3. Temperament

Temperament was evaluated using the Spanish version of the Children’s Behavior Questionnaire Short Form (CBQ-SF) [58]. The test is completed by parents, who evaluate their children’s reactions to different situations. It comprises 94 items (for example: usually rushes into an activity without thinking about it, is not very bothered by pain) on seven-point Likert-type scale (from “extremely untrue of your child” to “extremely true of your child”).

The questionnaire provides information on 15 dimensions of temperament: activity level, anger/frustration, positive anticipation, attentional focusing, discomfort, falling reactivity and soothability, fear, high intensity pleasure, impulsivity, inhibitory control, low intensity pleasure, perceptual sensitivity, sadness, shyness and smiling/laughter. One of the dimensions of the questionnaire (positive anticipation) did not have enough reliability in our sample and was therefore eliminated during the subsequent analyses.

To sort the temperamental dimensions into groups, factorial analyses were carried out using the varimax rotation. The results revealed three main factors:-Factor 1 (surgency): activity level, high intensity pleasure, impulsivity, shyness (negative);-Factor 2 (effortful control): attentional focusing, inhibitory control, low intensity pleasure, perceptual sensitivity, smiling/laughter;-Factor 3 (negative emotionality): anger/frustration, discomfort, falling reactivity and soothability (negative), fear, sadness.

### 2.4. Parenting Styles

Father’s parenting styles were measured using the Parenting Styles and Dimensions Questionnaire (PSDQ) [59]. This questionnaire evaluates the frequency with which parents engage in certain behaviors in relation to their children, in order to identify their parenting style. It measures the authoritative, authoritarian and permissive styles and in our study was administered to the fathers. It consists of 62 items (for example, I show sympathy when our child is hurt or frustrated, I am easygoing and relaxed with our child), with a four-level Likert-type response scale (never, occasionally, very often, and always).

The authoritative scale is defined by four subscales: warmth/involvement, reasoning/induction, democratic participation and good natured/easygoing. The authoritarian scale is also divided into four subscales: verbal hostility, corporal punishment, nonreasoning/punitive strategies and directiveness. Finally, the permissive parenting scale comprises three subscales: lack of follow-through, ignoring misbehavior and self-confidence. The reliability values of the scales in our sample were α = 0.86; α = 0.63, and α = 0.67 (respectively). High authoritarian and permissive and low authoritative parenting rates would be treated as an adverse family environment, while high authoritative and low authoritarian and permissive parenting would be considered a favorable environment.

### 2.5. Hormone Level Measures

To analyze baseline cortisol levels, two saliva samples were collected by passive drool within a six-week timeframe. Samples were collected in the classrooms at 09.00 h in order to avoid concentration changes due to diurnal hormone fluctuations.

To facilitate salivation, participants were given a sweet but were told they were not allowed to eat it until they had given a large enough sample. Samples were processed in the laboratory of the University of the Basque Country and stored at −80° until analysis. The assays were carried out in duplicate for each subject using an ELISA (enzyme-linked immunosorbent assay) technique (Salimetrics, State College, PA, USA). The intra-assay coefficient of variation (CV) for cortisol was less than 3% and the inter-assay coefficient of variation (CV) was less than 14%, calculated on the basis of control samples.

In order to establish a baseline for each child’s cortisol levels, a Pearson correlation was performed to explore the relationship between the two saliva samples’ hormone levels. Due to the positive correlation (r = 0.560, *p* = 0.0001), the two values were averaged, and it was this value that was used for the statistical analyses.

### 2.6. Statistical Analyses

All the variables were transformed into Z scores in order to cancel the effect of the range disparity problems. As the variables did not follow a normal distribution, they were normalized using the Blom transformation, which is one of the best transformations for dealing with asymmetric distributions [60], since it mitigates the violation of the normality assumption.

First of all, to test for sex differences in all the variables studied, we conducted an ANOVA. Subsequently, to test for possible associations between the variables analyzed in the study, a Pearson correlation was performed.

Secondly, linear regression analyses were conducted to study the interaction effects of the potential moderator and predictive variables on aggressive behavior. A total of nine regression models were analyzed in which the three parenting styles (authoritative father, authoritarian father and permissive father) and the three temperamental factors (negative emotionality, surgency and effortful control) were introduced separately. In every model, the predictive variables were sex, cortisol, a parenting style, a temperamental factor, the two-way interactions and the three-way interaction. All the variables entered into the regression models were continuous except for sex. Analyses were performed using the SPSS 24.0 statistical package.

Finally, to evaluate which theoretical model (differential susceptibility or diathesis-stress) best explained the interactions, the technique described by Roisman, Newman, Fraley, Haltigan, Groh and Haydon [61] was followed using a freely available web-based program developed by Chris Fraley [62] http://www.yourpersonality.net/interaction/ (access date 24 October 2018). To this end, we analyzed the regions of significance (RoS), probing the interactions within a range of ±2 SD from the mean of the predictive variables (parenting styles) and the proportion of interaction (PoI).

According to Roisman et al. [61] the prototypic PoI values would be 50% for differential susceptibility and 0% for diathesis-stress.

The region of significance (RoS) defines a specific value of z at which the regression of y to x becomes significant. The proportion of interaction (PoI) is the proportion of the total area between the lines of an interaction plot. The PA index refers to the proportion of cases differentially affected by the crossover interaction. In other words, it refers to the proportion of cases in which the association between aggressive behavior and parenting style is “reversed or qualified” [61]. In this case, at least 16% of the cases must lie above the crossover point at which the regression lines intersect for the data to be considered congruent with a differential susceptibility model [61].

## 3. Results

### 3.1. Sex Differences, Means and Standard Deviations

Analyses of variance (ANOVA) were performed to analyze sex differences in all the study variables. The results revealed that boys scored higher in aggressive behavior (*F*(1,158) = 4.17, *p* < 0.043, *d* = 0.32) than girls. No statistically significant differences were found between boys and girls in any of the other variables (Table 1).

### 3.2. Correlation Analyses

Pearson correlations were carried out to study the relationship between aggressive behavior, cortisol levels, temperament and parenting styles. As shown in Table 2, aggressive behavior was found to correlate positively with surgency, negative emotionality and authoritarian father. Effortful control was positively correlated with authoritative father and permissive father. Finally, negative emotionality was found to correlate negatively with cortisol levels.

### 3.3. The Predictive Role of the Interaction between Moderators (Sex, Cortisol Levels and Temperament) and Parenting Styles in Relation to Aggression

Of all the regression models performed, only three revealed two or three-level interactions between the variables under study. As shown in Table 3 and Table 4, the model containing authoritative father and negative emotionality (*R*^2^ = 0.25; *F*(12,139) = 3.529; *p* = 0.0001) and the model containing authoritative father and effortful control (*R*^2^ = 0.20; *F*(12,139) = 2.716; *p* = 0.003) were found to be significant. In both models, a statistically significant interaction was observed between sex, cortisol and authoritative father for explaining aggressive behavior.

We conducted a “regions of significance” (RoS) analysis on the moderator (Z) to analyze the entire range of moderator (cortisol) values for which parenting style and aggressive behavior were significantly associated. The regression of aggression on authoritative father was statistically significant (alpha < 0.05) for all the cortisol values that fell outside the region (−6.458, 0.460).

Additional analyses for evaluating differential susceptibility [61] indicated that the crossover point of the simple slopes on parenting was 1.065, within the range of ±2 SD from the mean of authoritative father.

The proportion of the interaction (PoI) below the crossover point was 0.77, and the proportion above the crossover point was 0.23.

This indicates that the crossover interaction in Figure 1 is consistent with the diathesis-stress theory as posited by Roisman et al. [61], in which boys with high levels of cortisol have the highest levels of aggressive behavior when exposed to low-quality parenting (low authoritative father values).

The model containing permissive father and negative emotionality was also found to be significant (*R*^2^ = 0.264; *F*(12,139) = 3.805; *p* = 0.0001) (Table 5). A statistically significant interaction was observed between sex, negative emotionality and permissive father for explaining aggressive behavior.

We conducted a “regions of significance” (RoS) analysis on the moderator (Z) to analyze the entire range of moderator (negative emotionality) values for which parenting style and aggressive behavior were significantly associated. The regression of aggression on permissive father was statistically significant (alpha < 0.05) for all the negative emotionality values that fell outside the region (−1.358, 0.464).

Additional analyses for evaluating differential susceptibility [61] indicated that the crossover point of the simple slopes on parenting was −0.677, within the range of ±2 SD from the mean of permissive father.

The proportion of the interaction (PoI) below the crossover point was 0.09, and the proportion above the crossover point was 0.91.

This indicates that the crossover interaction in Figure 2 is consistent with the diathesis-stress theory posited by Roisman et al. [61], in which girls with high levels of negative emotionality have the highest level of aggressive behavior when exposed to low-quality parenting (high permissive father values).

## 4. Discussion

The main aim of this study was to determine whether variations in family context (adverse–favorable fathers parenting style) differentially predict aggressive behavior in eight-year-old children in accordance with temperament, cortisol levels and sex, in accordance with either the diathesis-stress or the differential susceptibility model [9,63]. Our results support the idea of a certain biological predisposition, which renders children brought up in unfavorable contexts more vulnerable and causes them to display more aggressive behavior. These findings coincide with those reported by Ferguson and Dick [5] and Swearer and Hymel [6], within the diathesis-stress model.

Following the established hypotheses, the results revealed that girls with high negative emotionality and permissive fathers showed high levels of aggressive behavior. This is consistent with the diathesis-stress model, following the criterion proposed by Roisman et al. [61], which posits that girls with high levels of negative emotionality (as a vulnerability factor) are more affected by a negative environment, in this case a permissive father, and that this contributes to explaining their high levels of aggression. The finding is also consistent with results reported by other authors [11,64], who observed that negative emotionality and family environment were linked to aggression levels among children. In a recent study, Ren and Zhang [23] found that children’s negative emotionality and a negative paternal parenting style were linked to externalizing problems in child behavior. This study also highlighted the need to analyze other aspects of negative emotionality, in addition to irritability (the only variable studied) and to explore parenting practices in more detail. In our study, one plausible interpretation of this finding is that daughters with high levels of anger, sadness, fear and distress (high negative emotionality) perhaps do not have a father figure that shows firm and sets limits to their difficult temperamental profile, what makes them more aggressive. Indeed, recent studies suggest that fathers’ permissiveness may have a particularly strong influence on their daughters’ aggressive behavior [56,65] which may explain why we found this association among girls but not among boys. It seems clear that negative emotionality may be considered a vulnerability marker.

As regards the second hypothesis established in the study, boys with high cortisol levels were found to engage in more aggressive behavior when they were brought up in an environment with low levels of paternal authoritative parenting style (which could be understood as an adverse context). This finding is consistent with the diathesis-stress model, in accordance with the criteria established by Roisman et al. [61].

In relation to cortisol, an association has been found between the unregulated response of this hormone to psychosocial stressors and different psychopathologies in children, adolescents and adults [66,67,68]. Results have also been reported which link both cortisol hyporeactivity [69] and cortisol hyperreactivity [37,70] with externalizing behavior. Thus, it has been suggested that children with abnormal stress reactivity (measured through cortisol) are at risk of engaging in maladaptive behavior when exposed to adverse environments, including negative parenting styles and high levels of family stress [9,71,72,73]. Moreover, some studies seem to support the idea that this combination, i.e., high cortisol reactivity and adverse context associated with externalizing behavior, occurs only among boys [45,46]. Thus, in a study by Hastings, Ruttle, Serbin, Mills, Stack and Schwartzman [74], the authors found that the combination of an increase in cortisol reactivity and high levels of family stress were associated with more externalizing problems in boys, but not in girls.

It is important to emphasize the fact that, to date, no study has considered baseline levels of cortisol as a vulnerability factor. The present study, therefore, broadens the field of research in relation to this hormone, viewing it as a vulnerability element in the aforementioned models and analyzing not just reactivity levels, but circulating baseline levels also. These baseline levels are linked to the regulation of diverse biological processes, such as energy release, immune activity and mental activity, among others [75], which help the body adapt to the demands of its environment [29]. It is, therefore, very important to include them in the study of susceptibility or vulnerability models, particularly those that aim to explain aggressive behavior.

In sum, our results support the idea of certain biological and temperamental vulnerability characteristics, such as high cortisol levels and negative emotionality. These characteristics render both boys and girls more vulnerable to adverse family contexts, in this case understood as either a permissive or not particularly authoritative parenting style. Adverse contexts have a stronger negative effect on these children, making them more aggressive, although it should be remembered that the level of aggression analyzed in this study was non-pathological rather than severe or clinical and that our context has a limited range of adversity and stress. It is also important to highlight the need for a statistical tool capable of analyzing two continuous moderator variables, as temperament and cortisol, while at the same time exploring them in accordance with the diathesis-stress model or differential susceptibility theory. That would bring the opportunity to observe whether these two variables jointly act as a vulnerability or susceptibility factors and if they have a cumulative effect.

## 5. Conclusions

The findings of this study highlight the importance of the father’s parenting style, which seems to have a greater influence than originally thought. The majority of research into associations between childrearing and aggressive behavior among children have focused on mothers, since it has generally been assumed that mothers spend more time engaged in childcare [76]. Nevertheless, studies such as the one presented here show that the role of the father in childrearing is fundamental to understanding children’s aggressive behavior and highlight the need to pay special attention to psychobiological factors such as hormone levels and temperament as elements of vulnerability that may be interesting for planning future preventive practices.

However, it must be taken into account that this study was carried out with a not too large sample, so they should be replicated with a larger sample in future studies where other relevant hormones, as testosterone, could also be included in the study of aggressive behavior.

## Figures and Tables

**Figure 1 brainsci-11-01088-f001:**
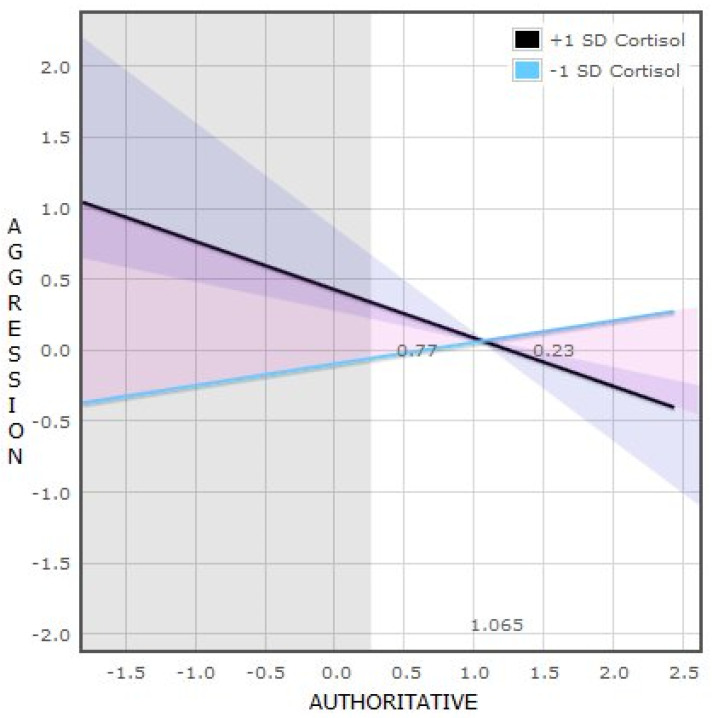
Interaction between cortisol levels and authoritative father in relation to aggressive behavior in boys.

**Figure 2 brainsci-11-01088-f002:**
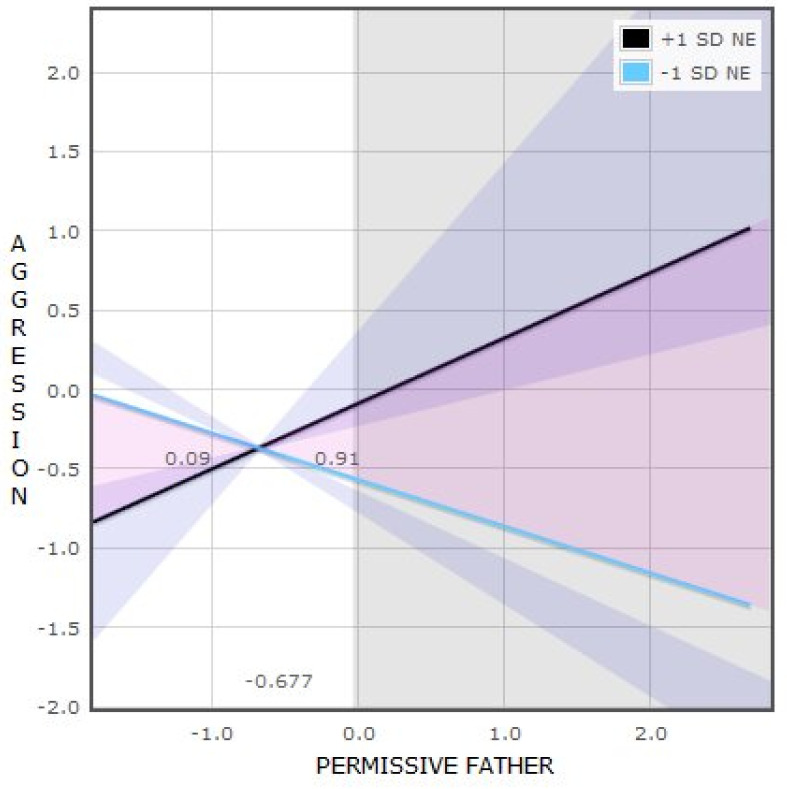
Interaction between negative emotionality and permissive father in relation to aggressive behavior in girls.

**Table 1 brainsci-11-01088-t001:** Descriptive statistics and analyses of variance.

Variable	Range	Boys (*n* = 88)		Girls (*n* = 70)		F
		M	SD	M	SD	
Aggressive Behavior	0–3	0.64	0.39	0.52	0.35	4.175 *
Cortisol (µg/dL)	0.04–0.79	0.283	0.133	.2886	0.155	0.5
Surgency	1.6–4.97	3.260	0.618	3.221	0.482	0.182
Effortful Control	1.6–6.29	5.012	0.574	5.222	0.797	3.583
Negative Emotionality	1.6–5.9	3.969	0.630	4.096	0.649	1.503
Authoritative Father	0–3.98	2.867	0.799	2.879	0.915	0.007
Authoritarian Father	0–2.67	1.878	0.530	1.821	0.586	0.377
Permissive Father	0–2.49	1.677	0.487	1.628	0.549	0.318

* *p* < 0.05.

**Table 2 brainsci-11-01088-t002:** Correlations between study variables.

	1	2	3	4	5	6	7	8
1. Aggression	1							
2. Surgency	0.239 **	1						
3. Effortful Control	−0.097	−0.178 *	1					
4. Negative Emotionality	0.179 *	−0.064	−0.031	1				
5. Cortisol	0.254 **	−0.016	0.122	−0.182 *	1			
6. Authoritative Father	−0.083	−0.144	0.250 **	0.019	−0.048	1		
7. Authoritarian Father	0.172 *	0.038	−0.074	0.073	−0.100	0.236 **	1	
8. Permissive Father	0.058	0.021	−0.096	0.156	−0.031	0.021	0.474 **	1

* *p* < 0.05. ** *p* < 0.01.

**Table 3 brainsci-11-01088-t003:** Regression analysis for aggressive behavior including authoritative father and negative emotionality.

	β	t	p
Sex	−0.200	−2.555	0.012 *
Cortisol Levels	−0.431	−1.750	0.082
Negative Emotionality	0.305	1.258	0.211
Authoritative Father	−0.183	−0.766	0.445
Sex * Cortisol	0.736	2.961	0.004 **
Sex * Negative Emotionality	−0.032	−0.133	0.895
Sex * Authoritative Father	0.122	0.507	0.613
Cortisol * Negative Emotionality	−0.019	−0.232	0.817
Cortisol * Authoritative Father	−0.615	−2.317	0.022 *
Authoritative Father * Negative Emotionality	−0.151	−0.618	0.537
Sex * Cortisol * Authoritative Father	0.547	2.037	0.044 *
Sex * NE * Authoritative Father	0.185	0.755	0.452

* *p* < 0.05. ** *p* < 0.01.

**Table 4 brainsci-11-01088-t004:** Regression analysis for aggressive behavior including authoritative father and effortful control.

	β	t	p
Sex	−0.124	−1.457	0.148
Cortisol Levels	−0.449	−1.761	0.081
Effortful Control	−0.048	−0.175	0.861
Authoritative Father	−0.095	−0.352	0.725
Sex * Cortisol	0.668	2.599	0.010 *
Sex * Effortful Control	0.034	0.122	0.903
Sex * Authoritative Father	0.076	0.281	0.779
Cortisol * Effortful Control	0.095	1.019	0.310
Cortisol * Authoritative Father	−0.793	−2.737	0.007 **
Authoritative Father * Effortful Control	0.395	1.447	0.150
Sex * Cortisol * Authoritative Father	0.649	2.315	0.022 *
Sex * EC * Authoritative Father	−0.280	−0.995	0.322

* *p* < 0.05. ** *p* < 0.01.

**Table 5 brainsci-11-01088-t005:** Regression analysis for aggressive behavior including permissive father and negative emotionality.

	β	t	p
Sex	−0.255	−3.223	0.002 **
Cortisol Levels	−0.398	−1.590	0.114
Negative Emotionality	0.385	1.596	0.113
Permissive Father	−0.375	−1.507	0.134
Sex * Cortisol	0.706	2.822	0.006 **
Sex * Negative Emotionality	−0.102	−0.418	0.677
Sex * Permissive Father	0.342	1.375	0.172
Cortisol * Negative Emotionality	0.009	0.106	0.916
Cortisol * Permissive Father	0.184	0.632	0.529
Permissive Father * Negative Emotionality	−0.442	−1.872	0.064
Sex * Cortisol * Permissive Father	−0.120	−0.415	0.679
Sex * NE * Permissive Father	0.570	2.370	0.019 *

* *p* < 0.05. ** *p* < 0.01.

## Data Availability

The data presented in this study are available on request from the corresponding author. The data are not publicly available due to the Ethics Committee of the University to which authors belong.

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
