# Peer review of "Differential Susceptibility or Diathesis-Stress: Testing the Moderating Role of Temperament and Cortisol Levels between Fathers’ Parenting and Children’s Aggressive Behavior"

_brainsci, 2021, doi:10.3390/brainsci11081088_

Round 1
Reviewer 1 Report
This is an interesting and well-written paper, which adds significantly to the body of knowledge on aggressive behaviour in boys.
However, I have two qualifications in recommending publication. It would have been easy to include mothers in the study, and a fuller explanation of why they were not included would be welcome. Adding girls as subjects would also have enriched the study, since the variable of family context could have been included in the analyses (eg hi or lo cortisol in one or both parents; with differential effects on gender of child etc). Perhaps the authors can advocate a more complex design in their next study.
The second qualification concerns the use of Spanish-language versions of English-language (usually American) scales. Can the authors reference any work on scale translation (e.g. use of back-translation); and any validity studies of these translated instruments?
Author Response
Dear reviewer 1,
Thank you very much for your review. Hope that you consider our responses to your comments apropiated.
- The objective of the present work was to study the figure of the father; therefore, the mothers were not studied. As explained in lines 118-125 of the introduction, most studies have only focused on the role of the mother, and although there are more and more studies that include fathers in their samples, they are still very few who only focus on the father figure. This type of study can help to better understand the role of the father, by himself, without the interference of the mother.
- Regarding the presence of girls in this study, we have to clarify that girls constitute almost half of the sample in this study. As explained in the “participants” section (line 140), the sample consisted of 88 boys and 70 girls. In addition, in the results it can be observed how the analysis are made with boys and girls included (sex based interaction included in regression analysis), but the analysis of the significant interactions of the regressions (which included sex) showed that the results of this study were only significant in boys.
- In the section “Materials and Methods” (line 139) we have referenced for each scale the works for the properties of the scales used.
Reviewer 2 Report
This is a very interesting paper devoted to testing the moderating role of temperament and cortisol levels between fathers’ parenting and children’s aggressive behavior.
In my opinion the phenomenon analyzed by Authors is very complex, so the results should be treated as preliminary and conclusions drawn cautiously. The study group is relatively low (158) and thus the interpretation of data must be very careful.
In general the study design, methodology, and analysis of data are done properly.
But do the Authors plan any follow-up study on a larger sample? Perhaps a multicenter experiment could be a good option. And maybe some other hormones should be taken into consideration like e.g. testosteron.
Did the Bioethics Committee approve the following procedure? For me it looks a little bit like a manipulation:
"To facilitate salivation, participants were given a sweet but were told they were not allowed to eat it until they had given a large enough sample."
I do not feel and expert in English language, but I think some grammatical corrections are needed, eg. "Most research have only focused", shouldn`t it be "Most research has only focused". Please, consult it with an English language specialist of a native speaker.
Author Response
Dear reviewer 2,
Thank you very much for your review. Hope that you consider our responses to your comments apropiated.
- Considerations regarding the small size of our sample and future research that may include more hormones, such as testosterone, have been added to the conclusions and future direction section (line 410-413).
-
Yes, the ethics committee approved the procedure described since the candy did not interfere with the saliva samples, it only helped the children to salivate better when thinking about the candy (they could not eat it until they had finished collecting the samples saliva). At these ages, it is very normal to find children with difficulties when it comes to producing and spitting up the necessary amount of saliva. The presence of the candy helps to relax that moment and to motivate children to spit more and in less time.
-
We have changed the verb "have" to "has", and a native speaker has reviewed the manuscript in order to detect and correct possible grammatical errors.